# Dengue Fever Complicated with Hemophagocytic Lymphohistiocytosis: A Case Report of Resolution with Steroid-Sparing Supportive Care

**DOI:** 10.3390/tropicalmed8110497

**Published:** 2023-11-15

**Authors:** Naoya Mizutani, Tsuneaki Kenzaka, Hogara Nishisaki

**Affiliations:** 1Department of Internal Medicine, Hyogo Prefectural Tamba Medical Center, Tamba 669-3495, Japan; nmizutani-hyg@umin.ac.jp (N.M.); honssk-d@sanynet.ne.jp (H.N.); 2Division of Community Medicine and Career Development, Kobe University Graduate School of Medicine, Chuo-ku, Kobe 650-0017, Japan

**Keywords:** dengue fever, thrombocytopenia, hemophagocytic lymphohistiocytosis, corticosteroids, supportive care

## Abstract

Dengue fever (DF) can be complicated by hemophagocytic lymphohistiocytosis (HLH). Steroid administration is markedly effective for this hematologic complication, and for other viral infections. We present a rare case of DF-associated HLH that improved with steroid-sparing supportive care. A 47-year-old Japanese male with diabetes mellitus and no history of DF traveled to the Philippines 10 days before his hospitalization. Three days before emergency admission, he experienced fever and joint pain and was referred to our hospital for suspected DF, after blood tests indicated liver damage and thrombocytopenia. Erythema of the extremities and trunk appeared on day 2, and the next day neutrophils were 550 cells/μL, platelets 29,000 cells/μL, ferritin 9840 ng/mL, and fibrinogen 141 mg/dL. Bone marrow aspirate revealed hemophagocytic lymphohistiocytosis, and he was diagnosed with HLH. On day 4, the symptoms and findings improved; only supportive care without steroids was continued. He tested positive for dengue virus antigen on admission. He was discharged on day 9 of hospitalization in good general condition with no vascular leakage or bleeding and recovery of blood cells. Although steroid administration is markedly effective in cases of DF complicated by HLH, this case suggests that such cases can resolve with steroid-sparing supportive care.

## 1. Introduction

Dengue fever (DF) is an infectious disease caused by the dengue virus, which is transmitted by striped dengue and human striped dengue mosquitoes. The disease is most prevalent in tropical and subtropical regions and is estimated to affect approximately 100 million people worldwide annually [1]. Its onset includes symptoms such as fever, headache, myalgia, and joint pain, accompanied by a skin rash that begins in the trunk 3 to 4 days after onset. Most cases resolve spontaneously in about a week with no complications. However, infants, older adults, pregnant women, diabetics, and patients with renal failure have an increased risk of severe disease, which may even lead to death [2].

In DF, the presence of warning signs such as persistent vomiting, abdominal pain, hepatomegaly, elevated hematocrit, ascites/pleural effusion, mucosal hemorrhage, thrombocytopenia, and lethargy increases the risk of serious disease and can lead to dengue hemorrhagic fever or dengue shock syndrome [2].

Mechanisms of thrombocytopenia in DF include reported impaired hematopoiesis due to infection of hematopoietic cells in the bone marrow [3], and immunological mechanisms due to the appearance of antiplatelet antibodies [4]. Other possible mechanisms include hemophagocytic lymphohistiocytosis (HLH). Combined cases of DF and HLH can have fatal outcomes, with 8 of the 27 cases reported from 1996 to 2014 resulting in death [5]. There are several case reports of significant efficacy of steroid administration in DF with HLH [6,7]. However, there are no reports that fully demonstrate its efficacy. We present a case of DF complicated by HLH that was resolved with steroid-sparing supportive care.

## 2. Case Description

A 47-year-old Japanese man, with well-controlled diabetes mellitus by dietary intervention alone and a history of reflux esophagitis, travelled to the suburb of Manila, Philippines, 10 days prior to his hospitalization. At the time, DF was not prevalent in Japan, and there was no history of DF. He did not recall any mosquito bites during his stay. He used insect repellents, but not DEET (N,N-diethyl-3-methylbenzamide) or highly effective ones such as icaridin. He did not consume raw water or unheated foods. He received the influenza vaccine 4 days prior to admission. Three days before admission, he became aware of fever and joint pain. The next day, his physician diagnosed an adverse reaction to the influenza vaccine and placed him under observation. Subsequently, he continued to have fever above 40 °C, headache, and fatigue, which were also noted on the day of hospital admission. His laboratory tests revealed liver damage (aspartate aminotransferase (AST) 357 U/L; alanine aminotransferase (ALT) 242 U/L) and low platelets (73,000/µL), so he was referred to our hospital for further examination.

On arrival to the hospital, his consciousness was clear. His body temperature was 39.8 °C, blood pressure was 109/65 mmHg, pulse rate was regular at 95 beats per minute, respiratory rate was 20 breaths per minute, and oxygen saturation was 97% while breathing ambient air. On physical examination, he had hyperemia of the eyelid conjunctiva and mild tenderness in the orbital area. Enlarged lymph nodes in the right posterior neck were palpable. The tourniquet test was negative. Blood tests revealed white blood cell counts of 2500 cells/µL (neutrophils 68.8%, lymphocytes 24.4%, monocytes 6%), hemoglobin 16.8 g/dL, hematocrit 47.4%, and platelets 73,000/µL. Biochemistry showed elevated transaminases (AST 375 U/L, ALT 272 U/L), albumin 4.5 g/dL, and lactate dehydrogenase (LDH) 714 U/L (standard value 120–220 U/L). Immunoserology tests were negative for hepatitis A immunoglobulin M (HA-IgM) antibody, hepatitis B surface (HBs) antigen, hepatitis C virus (HCV), hepatitis E immunoglobulin A (HE-IgA), Epstein–Barr virus capsid antigen-immunoglobulin M and G (VCA-IgM and G), Epstein–Barr virus nuclear antigen immunoglobulin G (EBNA-IgG), cytomegalovirus immunoglobulin M and G (CMV-IgM and G), and herpes simplex virus immunoglobulin M and G (HSV-IgM and G) antibodies. Urinalysis showed no abnormal findings. Contrast-enhanced computed tomography of the abdomen showed a splenomegaly of 120 mm in length and hepatomegaly. Culture tests of blood, stool, urine, and sputum on admission were all negative. At that time, the number of DF cases in the Philippines had nearly doubled from the previous year. Therefore, DF was the most likely diagnosis, and typhoid fever was listed as the differential diagnosis. The patient was hospitalized for treatment. He was a diabetic at high risk of severe disease and showed warning signs [2] such as abdominal pain, hepatomegaly, and thrombocytopenia. Considering the possibility of transition to dengue hemorrhagic fever or dengue shock syndrome, intravenous fluid infusion was started at 3 L/day. Erythema of the extremities and trunk appeared on day 2 (Figure 1), and the next day the white blood cell count halved to 1250 cells/µL; his blood test also revealed neutrophils 550 cells/µL, hemoglobin 15.5 g/dL, hematocrit 43.6%, platelets 29,000/µL, fibrinogen 141 mg/dL, ferritin 9840 ng/mL, and soluble interleukin-2 receptor 982 U/L. HLH was suspected due to the presence of two lineages with cytopenia. During platelet transfusion, bone marrow aspiration and culture were performed to rule out typhoid fever. The bone marrow test showed hemophagocytosis and hyperplasia (Figure 2). The patient met the diagnostic criteria for HLH-2004 [8] as shown in Table 1 and was diagnosed with HLH. Bone marrow culture was negative for *Salmonella typhi*. On day 5, the tourniquet test was repeated again, and the result was negative. The patient was diagnosed with HLH, and we considered administering steroids; however, we decided not to do so, because of the patient’s overall condition; his general appearance, headache, myalgia, appetite and food intake, fever, and white blood cell counts all showed a trend toward improvement. Recovery of blood cells was observed, and the fever resolved. On day 8, the patient was confirmed positive for the dengue virus NS1 antigen, which was submitted at admission, and the diagnosis was confirmed as DF and dengue-associated HLH. His condition improved enough for discharge on day 9 (Figure 3).

On day 14, the platelet count increased to 298,000 cells/μL, and a follow-up outpatient examination was completed. The patient has been visiting the hospital regularly for dietary modification for diabetes mellitus, and, after one year, he is currently living a healthy life with no flare-ups of HLH.

## 3. Discussion

We present a rare case of DF complicated by HLH. This case suggests that DF can be managed with only steroid-sparing supportive care if the disease shows signs of improvement. HLH is a rare complication of DF. The HScore [9] can be used to estimate the risk of HLH complications in patients with DF (Table 2).

We evaluated and compared HScores at two timepoints: on admission and on day 3 before the bone marrow examination (Table 3). At the latter timepoint, the HScore was approaching its best cutoff value of 169 (sensitivity of 93%, specificity of 86%, and accurate classification of 90% of the patients) and fulfilled five of the criteria shown in Table 1. Since it was necessary to rule out typhoid fever because this case was a traveler’s infection, we performed a bone marrow examination as a diagnostic procedure, which increased the HScore by 35 to 188 and raised the probability of having HS or HLH to 78.0%.

The mechanisms underlying HLH remain unclear, but the currently accepted theory is that inappropriately proliferating and activated T cells cause macrophage activation, impairing intracellular killing of phagocytes [10].

Perforin is an important effector molecule of lymphocyte cytotoxicity and is involved in innate and adaptive immune responses [11]. Perforin protein, expressed on CD8+ T and natural killer (NK) cells, forms holes in target cell membranes and is essential for killing through non-Fas-mediated mechanisms [11]. NK cells constitutively express perforin and can rapidly kill host cells infected with viruses or derived from tumors. Cytotoxic T cells emerge as effectors of adaptive immunity because they regulate perforin expression upwards after T cell receptor activation [12].

Important roles for perforin and NK cells in the mechanisms underlying HLH have been suggested. Familial hemophagocytic lymphohistiocytosis (FHLH2), which is an inherited autosomal recessive disease, is characterized by mutations in the perforin coding sequence [12]. Patients with mutations in the perforin gene (*PRF1*) lack or have low levels of perforin in NK cells and reduced lymphocyte cytotoxicity [12].

Acquired or secondary HLH is associated with malignancy, autoimmunity, and infection (most commonly EBV) [13]. Patients with HLH present with a variety of other characteristic symptoms, including jaundice, lymphadenopathy, hepatomegaly, rashes, focal neurological deficits, and seizures. On further assessment, these patients are found to have markedly high levels of interleukin 10, interleukin 6, tumor necrosis factor-α, interferon gamma (IFN-γ), and macrophage colony-stimulating factor (M -CSF) [13].

The clinical manifestations of HLH are caused by a hyperinflammatory syndrome due to hypercytokinaemia of various inflammatory mediators [14]. Uncontrolled activation of immune cells is probably caused by an exogenous irritant such as infectious microorganisms or toxins, or by a disabling immune response caused by endogenous factors such as radical stress, tissue damage, or metabolic products [14].

The persistence of pathogen-infected cells results in sustained antigen stimulation, proliferation of T cells and macrophages, and elevated pro-inflammatory cytokines [15,16]. The impaired cytotoxicity of CD8+ T cells allows antigen-presenting cells to continue presenting antigens, thereby increasing the proliferation of these T cells. This leads to uncontrolled production of IFN-γ and other cytokines, resulting in the characteristic histological and clinical features of HLH, such as hemophagocytosis [17].

An observational study of 62 Thai children with dengue hemorrhagic fever/dengue shock syndrome aged 4–12 years and 59 normal controls of the same age [18] showed significantly increased per-cell NK cell activity in the early stages of disease compared with the late stages of disease and normal controls. This suggests that NK cells are active in defense against dengue virus infection and may play a role in the etiology of DF deterioration. Furthermore, invariant natural killer T (iNKT) cells have been shown to be activated during acute dengue infection [19]. The iNKT cells are multifunctional, as indicated by their ability to proliferate after stimulation and to secrete IFN-γ and GM-CSF [20].

Combined cases of DF and HLH may have a fatal outcome. It has been reported that of the 27 cases described in the literature between 1966 and 2014, 8 resulted in death [5]. HLH associated with DF is often treated with steroids, and there are reports of substantial responses to steroid administration [6,7]. A PubMed search using the terms “dengue fever” and “hemophagocytic lymphohistiocytosis” identified 28 cases of DF and HLH in patients 19 years of age and older, described in 18 case reports to date; these reports, together with the present case, are summarized in Table 4 [5,6,7,21,22,23,24,25,26,27,28,29,30,31,32,33,34,35]. The median age of the patients included in these reports was 34 years (range, 19–65), and the male-to-female ratio was 13 to 16. Dengue virus was type 1 in one case, type 2 in one case, and type 3 in four cases. Five of these twenty-nine cases died (17.2%); specifically, of the twenty patients who received steroids, three died (15.0%); of the eight patients who received only supportive care, one died (12.5%); and one patient died after receiving interferon.

There appeared to be no difference in mortality between the steroid and supportive care groups. As described in this case, after the acute phase, dengue infection is expected to improve. If a patient is in good general condition and DF has reached its acute stage and is improving, the patient’s condition can resolve with steroid-sparing supportive care, even if DF is complicated by HLH. Evaluating disease activity according to the clinical course of each case is important, especially when a patient is critically ill. Prospective case–control studies are needed to examine the efficacy of steroids in the future, although there are not many cases of DF complicated by HLH. A case series is expected to provide more definitive conclusions.

## 4. Conclusions

We present a rare case of DF complicated by HLH. Although steroid administration has been reported to be markedly effective in cases of HLH-complicated DF, our case suggests that DF complicated by HLH can resolve with steroid-sparing supportive care, if the disease shows signs of improvement.

## Figures and Tables

**Figure 1 tropicalmed-08-00497-f001:**
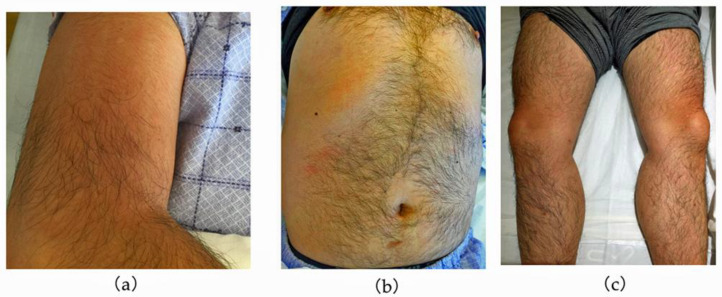
Physical examination findings: (**a**) right upper arm, (**b**) body core, (**c**) both lower limbs; erythema was present in each of these body parts.

**Figure 2 tropicalmed-08-00497-f002:**
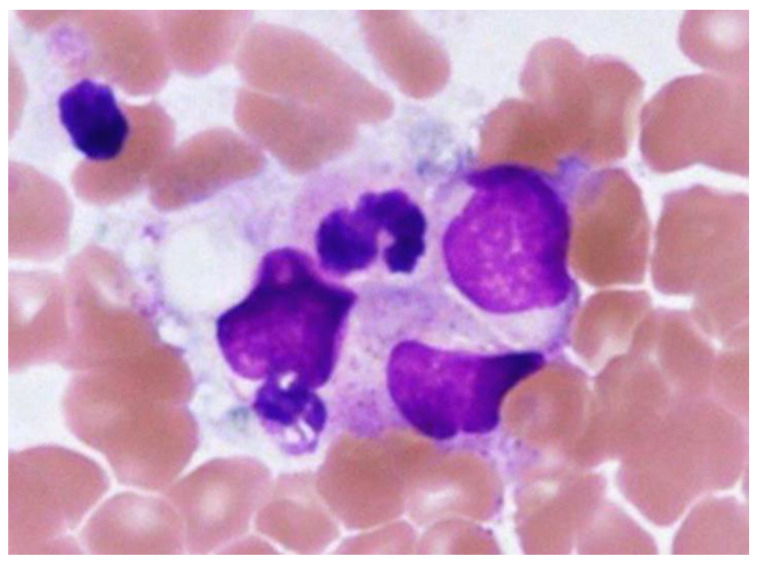
Bone marrow image. Hemophagocytosis and hyperplasia are observed.

**Figure 3 tropicalmed-08-00497-f003:**
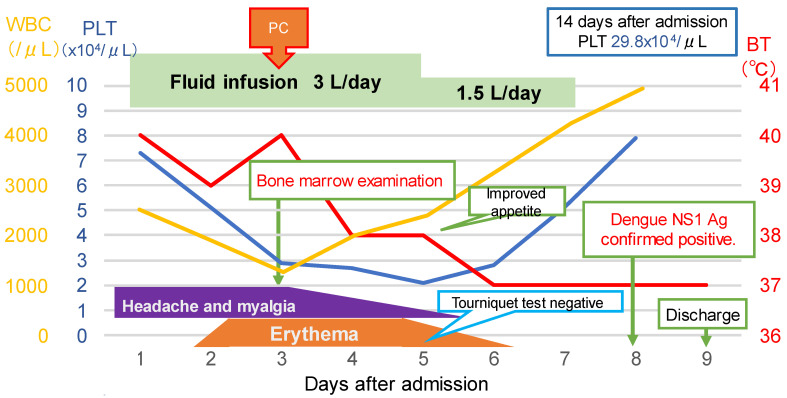
Clinical course. The red line shows the course of body temperature, the yellow line shows the course of WBC count, and the blue line shows the course of platelet count. WBC, white blood cells; PLT, platelets; BT, body temperature; PC, platelet concentrates; NS1, non-structural proteins 1; Ag, antigen.

**Table 1 tropicalmed-08-00497-t001:** Revised diagnostic guidelines for HLH, modified from reference [8].

The diagnosis of HLH can be established if one of either 1 or 2 below is fulfilled.	
(1) A molecular diagnosis consistent with HLH	
(2) Diagnostic criteria for HLH fulfilled (five out of the eight criteria below):	
(A) Initial diagnostic criteria (to be evaluated in all patients with HLH):	
✓ Fever	
✓ Splenomegaly	
✓ Cytopenias (affecting ≥2 of 3 lineages in the peripheral blood)	
Hemoglobin < 9.0 g/dL	
✓ Platelets < 100 × 10^3^/µL	
✓ Neutrophils < 1.0 × 10^3^/µL	
✓ Hypertriglyceridemia and/or hypofibrinogenemia	
Fasting triglycerides ≥ 265 mg/dL	
✓ Fibrinogen ≤ 150 mg/dL	
✓ Hemophagocytosis in bone marrow or spleen or lymph nodes	
No evidence of malignancy	
(B) New diagnostic criteria:	
Low or absent NK cell activity (according to local laboratory reference)	
✓ Ferritin ≥ 500 ng/mL	
Soluble interleukin-2 receptor ≥ 2.4 × 10^6^ U/L	

✓ indicates an item that meets these criteria in the reported patient. HLH, hemophagocytic lymphohistiocytosis; NK, natural killer.

**Table 2 tropicalmed-08-00497-t002:** HScore [9].

Parameter	Criteria for Scoring
Known underlying Immunosuppression *	0 (no) or 18 (yes)
Temperature (degrees of Celsius)	0 (<38.4), 33 (38.4–39.4), or 49 (>39.4)
Organomegaly	0 (no), 23 (hepatomegaly or splenomegaly), or 38 (hepatomegaly and splenomegaly)
No of cytopenias **	0 (1 lineage), 24 (2 lineages), or 34 (3 lineages)
Ferritin (ng/mL)	0 (<2000), 35 (2000–6000), or 50 (>6000)
Triglycerides (mmoles/L)	0 (<1.5), 44 (1.5–4), or 64 (>4)
Fibrinogen (gm/L)	0 (>2.5) or 30 (≤2.5)
Serum glutamic oxaloacetic transaminase (IU/L)	0 (<30) or 19 (≥30)
Hemophagocytosis features on bone marrow aspirate	0 (no) or 35 (yes)

* Human immunodeficiency virus-positive or receiving long-term immunosuppressive therapy (i.e., glucocorticoids, cyclosporine, azathioprine); ** Defined as a hemoglobin level of <9.2 gm/dL and/or a leukocyte count of <5000/mm^3^ and/or platelet count of <110,000/mm^3^.

**Table 3 tropicalmed-08-00497-t003:** HScore results on admission and on day 3 before bone marrow examination. **Probability calculation of having HS is available at**
https://saintantoine.aphp.fr/score/ (cited 3 November 2023).

Parameter	Result	
	On Admission	Day 3 before Bone Marrow Examination
Known underlying immunosuppression	0 (no)	0 (no)
Temperature (degrees of Celsius)	49 (>39.4)	49 (>39.4)
Organomegaly	38 (hepatomegaly and splenomegaly)	Unknown (not detected)
No of cytopenias	24 (2 lineages)	24 (2 lineages)
Ferritin (ng/mL)	Unknown (not detected)	50 (>6000)
Triglycerides (mmoles/L)	Unknown (not detected)	Unknown (not detected)
Fibrinogen (gm/L)	Unknown (not detected)	30 (≤2.5)
Serum glutamic oxaloacetic transaminase (IU/L)	19 (≥30)	Unknown
Hemophagocytosis features on bone marrow aspirate	Unknown (not detected)	Unknown (not yet detected)
HScore	130	153
Probability of having HS (%)	8.9	28.8

**Table 4 tropicalmed-08-00497-t004:** Reported cases of dengue infection with HLH.

Ref.	Year	Age (y)	Sex	Nationality	Epidemicin Country	Fever Duration (Days)	Hospital Visit since Fever Onset (days)	Infection Type	Virus Serotype	Underlying Diseases	Treatment for HLH	Outcome
[21]	1991	19	Female	Singapore	Yes	−	−	−	−	−	Symptomatic	Alive
[22]	1991	53	Male	China	Yes	−	−	Primary	3	−	Symptomatic	Alive
[23]	2005	33	Male	China	Yes	10	4	Primary	−	−	Symptomatic	Alive
[24]	2008	46	Female	Thailand	Yes	4	−	Secondary	2	None	MP, IVIg	Alive
[25]	2009	32	Female	Japan	No	7	3	Primary	−	None	Symptomatic	Alive
[26]	2011	24	Female	India	Yes	7	4	Primary	−	None	Dexa	Alive
[27]	2011	22	Female	Thailand	Yes	−	−	−	−	None	Dexa, MP, IVIg	Alive
[27]	2011	43	Female	Thailand	Yes	−	−	−	−	None	Dexa, MP, IVIg	Alive
[27]	2011	45	Female	Thailand	Yes	−	−	−	−	None	Dexa, IVIg	Died
[27]	2011	65	Male	Thailand	Yes	−	−	−	−	None	Dexa, IVIg	Died
[28]	2012	43	Female	Malaysia	Yes	6	6	Secondary	−	Diabetes	MP	Died
[28]	2012	34	Male	Malaysia	Yes	4	4	Primary	−	None	MP	Alive
[28]	2012	36	Female	Malaysia	Yes	7	7	Secondary	−	None	MP	Alive
[28]	2012	20	Male	Malaysia	Yes	4	4	Primary	−	G6PD deficiency	Symptomatic	Alive
[5]	2013	22	Female	India	Yes	7	−	Primary	−	None	Dexa	Alive
[29]	2014	44	Female	France	No	−	−	Primary	−	−	Steroid	Alive
[29]	2014	38	Male	France	No	≤14	3	Primary	1	−	Dexa, IVIg	Alive
[29]	2014	25	Female	France	No	≤7	−	Primary	−	−	MP	Alive
[30]	2014	21	Female	Belgium	No	3	3	Secondary	3	None	IVIg	Alive
[31]	2014	63	Female	USA	Yes	1	1	Primary	3	Crohn’s disease, CAD	IFN	Died
[6]	2015	32	Male	Malaysia	Yes	9	5	Primary	−	None	Dexa, IVIg	Alive
[6]	2015	19	Male	Malaysia	Yes	5	4	Secondary	−	None	Dexa	Alive
[32]	2015	26	Male	Pakistan	No	17	14	−	−	−	Symptomatic	Died
[33]	2016	34	Male	Japan	No	6	0	Primary	−	Cholesteatoma	MP	Alive
[7]	2017	33	Female	Korea	No	8	3	Primary	3	None	Dexa	Alive
[34]	2017	56	Male	China	Yes	8	6	−	−	−	MP	Alive
[34]	2017	35	Female	Malaysia	Yes	−	3	−	−	Obesity	MP	Alive
[35]	2017	44	Male	Bangladesh	Yes	≤18	4	−	−	None	Symptomatic	Alive
Present case	2020	47	Male	Japan	No	6	3	−	−	Diabetes, reflux esophagitis	Symptomatic	Alive

CAD, coronary heart disease; HLH, hemophagocytic lymphohistiocytosis; G6PD, glucose-6-phosphate dehydrogenase; MP, methylprednisolone; Dexa, dexamethasone; IVIg, intravenous immunoglobulin.

## Data Availability

All data generated or analyzed during this study are included in the published article.

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
