# Peer review of "Dengue Fever Complicated with Hemophagocytic Lymphohistiocytosis: A Case Report of Resolution with Steroid-Sparing Supportive Care"

_tropicalmed, 2023, doi:10.3390/tropicalmed8110497_

Round 1

Reviewer 1 Report

Comments and Suggestions for Authors

First of all, I would like to thank the authors for the effort to publish this interesting article.

While it is interesting given the low incidence of the disease, there are several points that make it need to be reviewed comprehensively. 

The first important point is the diagnosis. It is not clear with the criteria currently defined in the literature whether the patient really has HLH secondary to Dengue or not.

Fever, splenomegaly, cytopenia, hypertriglyceridemia, hypofibrinogenemia, and hemophagocytosis used to be enough for the diagnosis of HLH. These criteria were modified in 2004 and the Histiocyte Society added three more criteria: low or absent NK cell activity, hyperferritinemia, and high-soluble interleukin-2-receptor levels. The diagnosis should include five out of the previous eight criteria.

In this case and considering that the hepatosplenomegaly is at the high limit of normal, the number of criteria for secondary HLH in this case are met although neither triglyceride level nor NK activity is mentioned. Although demonstration of hemophagocytosis in bone marrow or lymph node is not mandatory, a cytologic image could be of value. 

If we are talking about the HScore score for Reactive Hemophagocytic Syndrome, the risk with the data provided ranges between 40% and 55%.

This point should be clarified in an exhaustive manner and this data should also be provided in the discussion if it finally meets the criteria. 

In the discussion, the pathophysiology of the disease is well reflected, although it should be more detailed, indicating the primary form.

With respect to the figures expressed in this article, we should speak of the laboratory limit of normality of the lactate dehydrogenase (LDH) figures since this changes according to the laboratory.

The graph provided, although it is very visual and provides a lot of information, should be clearer with the same range of colors so that the reader does not get lost and, in addition, some more photographs should be provided, either of the patient's clinic, pathological anatomy or imaging, since this is a case report and could provide a lot of information.

Author Response

First of all, I would like to thank the authors for the effort to publish this interesting article.
Response: Thank you for your thorough review.

While it is interesting given the low incidence of the disease, there are several points that make it need to be reviewed comprehensively.
Response: Thank you for your comments. We have corrected all the points you raised, as explained below.

The first important point is the diagnosis. It is not clear with the criteria currently defined in the literature whether the patient really has HLH secondary to Dengue or not.

Fever, splenomegaly, cytopenia, hypertriglyceridemia, hypofibrinogenemia, and hemophagocytosis used to be enough for the diagnosis of HLH. These criteria were modified in 2004 and the Histiocyte Society added three more criteria: low or absent NK cell activity, hyperferritinemia, and high-soluble interleukin-2-receptor levels. The diagnosis should include five out of the previous eight criteria.

In this case and considering that the hepatosplenomegaly is at the high limit of normal, the number of criteria for secondary HLH in this case are met although neither triglyceride level nor NK activity is mentioned. Although demonstration of hemophagocytosis in bone marrow or lymph node is not mandatory, a cytologic image could be of value.
Response: Thank you for pointing this out. To clarify the diagnostic rational of the described case, we have inserted Table 1 in the revised manuscript (Line 133):

Table 1. Revised diagnostic guidelines for HLH, modified from reference [8].

The diagnosis HLH can be established if one of either 1 or 2 below is fulfilled.

(1) A molecular diagnosis consistent with HLH.

(2) Diagnostic criteria for HLH fulfilled (five out of the eight criteria below):

(A) Initial diagnostic criteria (to be evaluated in all patients with HLH):

✓ Fever

✓ Splenomegaly

✓ Cytopenias (affecting ≧2 of 3 lineages in the peripheral blood)

Hemoglobin < 9.0 g/dL

✓ Platelets < 100 × 103/µL

✓ Neutrophils < 1.0 × 103/µL

✓ Hypertriglyceridemia and/or hypofibrinogenemia

Fasting triglycerides ≧265 mg/dL

✓ Fibrinogen ≦ 150 mg/dL

✓ Hemophagocytosis in bone marrow or spleen or lymph nodes

 No evidence of malignancy

(B) New diagnostic criteria:

Low or absent NK-cell activity (according to local laboratory reference)

✓ Ferritin ≧500 ng/mL

Soluble interleukin-2 receptor ≧2.4 × 106 U/L

✓indicates an item that meets these criteria in the reported patient. HLH, hemophagocytic lymphohistiocytosis; NK, natural killer

Also, we have provided a bone marrow image showing hemophagocytosis in Figure 2 (Line 131). Because of the insertion of Table 1, we have amended the number of the “Reported cases of dengue infection with HLH” table in the revised manuscript (Table 3, Line 237).

If we are talking about the HScore for Reactive Hemophagocytic Syndrome, the risk with the data provided ranges between 40% and 55%. This point should be clarified in an exhaustive manner and this data should also be provided in the discussion if it finally meets the criteria.
Response: Thank you for pointing this out. In response to your comment, we have added reference [9] in the revised manuscript, in order to discuss the HScore of this case:

[9] Fardet, L.; Galicier, L.; Lambotte, O.; Marzac, C.; Aumont, C.; Chahwan, D.; Coppo, P.; Hejblum, G. Development and validation of the HScore, a score for the diagnosis of reactive hemophagocytic syndrome. Arthritis Rheumatol. 2014, 66, 2613–2620. DOI:10.1002/art.38690.

Moreover, we have added the section shown below in the revised manuscript (Lines 149-162):

The HScore [9] can be used to estimate the risk of HLH complication in patients with DF (Table 2-1).

Table 2-1. HScore [9].

Parameter

Criteria for scoring

Known underlying

immunosuppression*

0 (no) or 18 (yes)

Temperature (degrees of Celsius)

0 (< 38.4), 33 (38.4-39.4), or 49 (> 39.4)

Organomegaly

0 (no), 23 (hepatomegaly or splenomegaly), or 38 (hepatomegaly and splenomegaly)

No of cytopenias**

0 (1 lineage), 24 (2 lineages), or 34 (3 lineages)

Ferritin (ng/mL)

0 (< 2,000), 35 (2,000-6,000), or 50 (> 6,000)

Triglycerides (mmoles/liter)

0 (< 1.5), 44 (1.5-4), or 64 (> 4)

Fibrinogen (gm/liter)

0 (> 2.5) or 30 (≦2.5)

Serum glutamic oxaloacetic

transaminase (IU/liter)

0 (< 30) or 19 (≧30)

Hemophagocytosis features
on bone marrow aspirate

0 (no) or 35 (yes)

*Human immunodeficiency virus positive or receiving long-term immunosuppressive therapy (i.e., glucocorticoids, cyclosporine, azathioprine)
**Defined as a hemoglobin level of < 9.2 gm/dL and/or a leukocyte count of < 5,000/mm3 and/or platelet count of < 110,000/mm3

and compared HScore at two timepoints: on admission and on day 3 before bone marrow examination (Table 2-2). At the latter timepoint, HScore was approaching its best cutoff value of 169 (sensitivity of 93%, specificity of 86%, and accurate classification of 90% of the patients) and fulfilled five of the criteria shown on Table 1. Since it was necessary to rule out typhoid fever because this case was a traveler’s infection, we performed bone marrow examination as a diagnostic procedure, which increased HScore by 35 to 188 and raised the probability of having HS or HLH to 78.0%.

Table 2-2. HScore results on admission and on day 3 before bone marrow examination.

Probability calculation of having HS is available at https://saintantoine.aphp.fr/score/ [cited 03 November 2023].

Parameter

Result

On admission

Day 3 before

bone marrow examination

Known underlying

immunosuppression

0 (no)

0 (no)

Temperature (degrees of Celsius)

49 (> 39.4)

49 (> 39.4)

Organomegaly

38 (hepatomegaly and splenomegaly)

Unknown (not detected)

No of cytopenias**

24 (2 lineages)

24 (2 lineages)

Ferritin (ng/mL)

Unknown (not detected)

50 (> 6,000)

Triglycerides (mmoles/liter)

Unknown (not detected)

Unknown (not detected)

Fibrinogen (gm/liter)

Unknown (not detected)

30 (≦2.5)

Serum glutamic oxaloacetic

transaminase (IU/liter)

19 (≧30)

Unknown (not shown)

Hemophagocytosis features

on bone marrow aspirate

Unknown (not detected)

Unknown (not yet detected)

HScore

130

153

Probability of having HS (%)

8.9

28.8

Because of the insertion of Table 2-1 and Table 2-2, we have amended the number of the “Reported cases of dengue infection with HLH” table in the revised manuscript (Table 3, Line 237).

In the discussion, the pathophysiology of the disease is well reflected, although it should be more detailed, indicating the primary form.

Response: Thank you for pointing this out. We have added the following text in the Discussion section of the revised manuscript, in order to describe the pathophysiology of the disease in more detail (Lines 168-206):

Perforin is a critical effector molecule of lymphocyte cytotoxicity, involved in the innate and adaptive immune response [11]. It is a protein expressed in CD8+ T cells and natural killer (NK) cells that forms holes on target cell membranes and is essential for killing via non-Fas-mediated mechanisms [11]. NK cells express perforin constitutively, allowing the host to rapidly kill virus-infected and tumor-derived host cells. Cytotoxic T-cells upregulate perforin expression after T-cell receptor activation, and thus emerge as effectors of adaptive immunity [12].

Important roles for perforin and NK cells in the mechanism underlying HLH have been suggested. Familial hemophagocytic lymphohistiocytosis (FHLH2) is an inherited autosomal recessive disorder, characterized by mutations identified in the perforin coding sequence [12]. Patients with mutations in the perforin gene (PRF1) have absent or low perforin levels in NK cells and diminished lymphocyte cytotoxicity [12].

Acquired or secondary HLH has been associated with infection (most commonly with EBV), malignancy, and autoimmunity [13]. Patients with HLH variably display several other characteristic features, such as hepatomegaly, jaundice, adenopathy, rash, seizures, and focal neurologic deficits. When evaluated, these patients have been found to have strikingly high serum levels of numerous cytokines, including interferon gamma (IFN-γ), tumor necrosis factor-α, interleukin 6, interleukin 10, and macro-phage-colony-stimulating factor (M-CSF) [13].

The clinical presentation of HLH is caused by a hyperinflammatory syndrome because of hypercytokinemia of various pro-inflammatory mediators [14]. The un-controlled activation of immune cells is probably caused by an ineffective immune response triggered by exogenous stimulants, such as infectious organisms and toxins, or by endogenous factors, such as radical stress, tissue damage, or metabolic products [14].

The persistence of pathogen-infected cells results in sustained antigenic stimulation, T-cell and macrophage proliferation, and elevation of proinflammatory cytokines [15, 16]. The impairment in the cytotoxicity of CD8+ T cells allows antigen-presenting cells to continue presenting antigens, thereby increasing the proliferation of these T cells. This results in uncontrolled production of IFN-γ as well as other cytokines, leading to the characteristic histologic and clinical features of HLH, including hemophagocytosis. [17].

An observational study [18] of 62 Thai children with dengue hemorrhagic fever/dengue shock syndrome aged 4-12 years and 59 age-matched normal controls showed that NK cell activity on a per-cell basis was significantly increased in the early disease stage when compared to the later stages of the disease and to normal controls. This suggests that NK cells are active in the defense against dengue viral infection, and may play a role in the pathogenesis of DF aggravation. Furthermore, invariant natural killer T (iNKT) cells are shown to be activated during acute dengue infection [19]. iNKT cells are polyfunctional, as demonstrated by their ability to proliferate and secrete IFN-γ and GM-CSF after stimulation [20].

Moreover, in relation to the above text, we have also added references [11], [13], [14], [15], and [16] in the revised manuscript:

[11] Russell, JH.; Ley, TJ. Lymphocyte-mediated cytotoxicity. Annu Rev Immunol. 2002, 20, 323–370. DOI:10.1146/annurev.immunol.20.100201.131730.

[13] Jordan, MB.; Hildeman, D.; Kappler, J.; Marrack, P. An animal model of hemophagocytic lymphohistiocytosis (HLH): CD8+ T cells and interferon gamma are essential for the disorder. Blood. 2004, 104, 735–743. DOI:10.1182/blood-2003-10-3413.

[14] Janka, GE.; Schneider, EM. Modern management of children with haemophagocytic lymphohistiocytosis. Br J Haematol. 2004, 124, 4–14. DOI:10.1046/j.1365-2141.2003.04726.x.

[15] Arnaout, RA. Perforin deficiency: fighting unarmed? Immunol Today. 2000, 21, 592.

[16] Moretta, L.; Moretta, A.; Hengartner, H.; Zinkernagel, RM. On the pathogenesis of perforin defects and related immunodeficiencies. Immunol Today. 2000, 21, 593–594.

With respect to the figures expressed in this article, we should speak of the laboratory limit of normality of the lactate dehydrogenase (LDH) figures since this changes according to the laboratory.

Response: Thank you for your advice. We have stated the LDH standard value (120-220 U/L) in the revised manuscript (Line 77).

The graph provided, although it is very visual and provides a lot of information, should be clearer with the same range of colors so that the reader does not get lost and, in addition, some more photographs should be provided, either of the patient's clinic, pathological anatomy or imaging, since this is a case report and could provide a lot of information.

Response: Thank you for your suggestions. In response to your comment, we have:

  1. added the following notes to the legend of Figure 3: “The red line shows the course of body temperature, the yellow line shows the course of WBC count, and the blue line shows the course of platelet count.” (Line 142).
  2. added images of skin findings during the patient’s physical examination in Figure 1 (Line 127).
  3. added images of the bone marrow in Figure 2 (Line 131).

Reviewer 2 Report

Comments and Suggestions for Authors

The study report a case of dengue with hemophagocytic lymphohostiocytosis. While it may be interesting that the patient recovered without steriods, this single case is insufficent to recommend any changes in clinical practice. Importantly, table 1 shows there is no advantage of steriod treatment or no treatment. Based on these comments, I would recommend the authors to not consider this case study as a study to change clinical practice. Note that most doctors would still recommend steriod treatment as this is the default practice.

Comments on the Quality of English Language

Quality of English is fine

Author Response

The study report a case of dengue with hemophagocytic lymphohistiocytosis. While it may be interesting that the patient recovered without steriods, this single case is insufficent to recommend any changes in clinical practice. Importantly, table 1shows there is no advantage of steriod treatment or no treatment. Based on these comments, I would recommend the authors to not consider this case study as a study to change clinical practice. Note that most doctors would still recommend steroid treatment as this is the default practice.

Response: Thank you for your advice. We would like to imply that this case may show the possibility of steroids-sparing treatment for DF complicated by HLH. As stated in Lines 221-224 of the revised manuscript, we would also like to emphasize that evaluating disease activity according to the clinical course of each case is important, especially when a patient is critically ill.

In response to your comment, we have made the following amendments to our manuscript:

  1. At Line 148 of the revised manuscript, we have changed the phrase “supportive care without steroids” to “steroids-sparing supportive care”.
  2. At Lines 14, 51, and 222 of the revised manuscript, we have revised the term “steroid-free” to “steroid-sparing”.
  3. At Line 52 of the revised manuscript, we have deleted the word “alone”, in order to reduce the emphasis we put on this case’s steroid-sparing treatment.
  4. In addition, we have amended the title of our manuscript by changing the phrase “Only with Supportive Care without Steroids” to “with Steroids-sparing Supportive Care,” which we have further highlighted in Lines 25-26 and 231 of the revised manuscript.

Finally, please note that because of the insertion of Table 1, Table 2-1, and Table 2-2, we have amended the number of the “Reported cases of dengue infection with HLH” table in the revised manuscript (Table 3, Line 237).

Reviewer 3 Report

Comments and Suggestions for Authors

Dear authors. Thank you for submitting this manuscript.

These are some of my suggestions to improve your manuscript. 

Line 48: cured to resolved 

Line 57: flu vaccine to influenza vaccine 

Line 57: The three days to Three days

Line 59: flu vaccine to influenza vaccine

Line 61: blood test to laboratory tests

Line 73 and 74: HA-IgM, HBs antigen, HCV, HE-IgA, VCA-IgM, 73 IgG, EBNA-IgG, CMV-IgM, IgG, HSV-IgM, or IgG antibodies. All need to be spelled out. 

Line 90: "The patient met the diagnostic criteria for HLH-2004": How? please elaborate in more detail. I think you should add histopathology pictures to support the HLA diagnosis. 

Line 91: culture was negative for typhoid fever to culture was negative for Salmonella typhi

Line 96: fever resolution to fever resolved. 

Line 100: "follow-up was terminated" needs rewording. 

Line 101: dietary therapy to dietary modification

Line 107: This case suggests that DF can improve only with supportive care without steroids if the disease shows signs of improvement. Based on one case this conclusion is an overreach. 

Line 124 and 125 is repeated again in 142 and 143: "In the case of DF, the NK cell activity increased significantly in the early stage 142 of the disease, but decreased significantly in the stages of febrile shock [13]. Furthermore, 143 invariant natural killer T (iNKT) cells are known to be activated during acute dengue in- 144 fection [14]."

Line 147: "patient can be cured with steroid-free supportive care" needs rewording. 

Comments on the Quality of English Language

Your manuscript will benefit from a good English editor. 

Author Response

Dear authors. Thank you for submitting this manuscript.
Response: Thank you for your thorough review.

These are some of my suggestions to improve your manuscript. 
Response: Thank you for your comments. We have replied to all your helpful suggestions, as explained below.

Line 48: cured to resolved 
Response: Thank you for your suggestion. We have corrected the indicated text (Line 51).

Line 57: flu vaccine to influenza vaccine 
Response: Thank you for your suggestion. We have corrected the indicated text (Line 61).

Line 57: The three days to Three days
Response: Thank you for your suggestion. We have corrected the indicated text (Line 61).

Line 59: flu vaccine to influenza vaccine
Response: Thank you for your suggestion. We have corrected the indicated text (Line 63).

Line 61: blood test to laboratory tests

Response: Thank you for your suggestion. We have corrected the indicated text (Line 65).

Line 73 and 74: HA-IgM, HBs antigen, HCV, HE-IgA, VCA-IgM, 73 IgG, EBNA-IgG, CMV-IgM, IgG, HSV-IgM, or IgG antibodies. All need to be spelled out. 
Response: Thank you for your suggestion. We have spelled out all the indicated abbreviations (Lines 77-82).

Line 90: “The patient met the diagnostic criteria for HLH-2004”: How? Please elaborate in more detail. I think you should add histopathology pictures to support the HLA diagnosis. 
Response: Thank you for your suggestion. In response to your comment, we have added the text “as shown in Table 1” (Line 99) in the revised manuscript. We have also inserted Table 1 in the revised manuscript (Line 133):

Table 1. Revised diagnostic guidelines for HLH, modified from reference [8].

The diagnosis HLH can be established if one of either 1 or 2 below is fulfilled.

(1) A molecular diagnosis consistent with HLH.

(2) Diagnostic criteria for HLH fulfilled (five out of the eight criteria below):

(A) Initial diagnostic criteria (to be evaluated in all patients with HLH):

✓ Fever

✓ Splenomegaly

✓ Cytopenias (affecting ≧2 of 3 lineages in the peripheral blood)

Hemoglobin < 9.0 g/dL

✓ Platelets < 100 × 103/µL

✓ Neutrophils < 1.0 × 103/µL

✓ Hypertriglyceridemia and/or hypofibrinogenemia

Fasting triglycerides ≧265 mg/dL

✓ Fibrinogen ≦ 150 mg/dL

✓ Hemophagocytosis in bone marrow or spleen or lymph nodes

 No evidence of malignancy

(B) New diagnostic criteria:

Low or absent NK-cell activity (according to local laboratory reference)

✓ Ferritin ≧500 ng/mL

Soluble interleukin-2 receptor ≧2.4 × 106 U/L

✓indicates an item that meets these criteria in the reported patient. HLH, hemophagocytic lymphohistiocytosis; NK, natural killer

Also, we have provided a bone marrow image showing hemophagocytosis in Figure 2 (Line 131). Because of the insertion of Table 1, we have amended the number of the “Reported cases of dengue infection with HLH” table in the revised manuscript (Table 3, Line 237).

Line 91: culture was negative for typhoid fever to culture was negative for Salmonella typhi
Response: Thank you for your suggestion. We have corrected the indicated text (Line 100).

Line 96: fever resolution to fever resolved. 
Response: Thank you for your suggestion. We have corrected the indicated text (Line 105).

Line 100: “follow-up was terminated” needs rewording.
Response: Thank you for your suggestion. We have amended the indicated text to “follow-up outpatient examination was completed” (Lines 136-137). 

Line 101: dietary therapy to dietary modification
Response: Thank you for your suggestion. We have corrected the indicated text (Lines 137-138).

Line 107: This case suggests that DF can improve only with supportive care without steroids if the disease shows signs of improvement. Based on one case this conclusion is an overreach. 
Response: Thank you for your advice. We would like to imply that this case may show the possibility of steroids-sparing treatment for DF complicated by HLH. As stated in Lines 226-227 of the revised manuscript, we would also like to emphasize that evaluating disease activity according to the clinical course of each case is important, especially when a patient is critically ill.

In response to your comment, we have made the following amendments to our manuscript:

  1. At Line 148 of the revised manuscript, we have changed the phrase “supportive care without steroids” to “steroids-sparing supportive care”.
  2. At Lines 14, 51, and 225 of the revised manuscript, we have revised the term “steroid-free” to “steroid-sparing”.
  3. At Line 52 of the revised manuscript, we have deleted the word “alone”, in order to reduce the emphasis we put on this case’s steroid-sparing treatment.
  4. In addition, we have amended the title of our manuscript by changing the phrase “Only with Supportive Care without Steroids” to “with Steroids-sparing Supportive Care,” which we have further highlighted in Lines 25-26 and 231 of the revised manuscript.

Line 124 and 125 is repeated again in 142 and 143: "In the case of DF, the NK cell activity increased significantly in the early stage 142 of the disease, but decreased significantly in the stages of febrile shock [13]. Furthermore, 143 invariant natural killer T (iNKT) cells are known to be activated during acute dengue in- 144 fection [14]."
Response: Thank you for your advice. We have deleted the repetitive text present in Lines 220-223 of the revised manuscript, and the following word “Therefore”.

Line 147: "patient can be cured with steroid-free supportive care" needs rewording. 
Response: Thank you for your advice. We have amended the indicated text to “the patient can be cured with steroids-sparing supportive care, even if DF is complicated by HLH” (Lines 225-226).

Round 2

Reviewer 2 Report

Comments and Suggestions for Authors

Authors have sufficiently addressed my comments

Author Response

Response: Thank you for your thorough review and comments.
